# Electrochemical Performance of Nitrogen Self-Doping Carbon Materials Prepared by Pyrolysis and Activation of Defatted Microalgae

**DOI:** 10.3390/molecules28217280

**Published:** 2023-10-26

**Authors:** Xin Wang, Lu Zuo, Yi Wang, Mengmeng Zhen, Lianfei Xu, Wenwen Kong, Boxiong Shen

**Affiliations:** 1Tianjin Key Laboratory of Clean Energy and Pollution Control, School of Energy and Environmental Engineering, Hebei University of Technology, Tianjin 300401, China; 18631351619@163.com (L.Z.); 202121303019@stu.hebut.edu.cn (Y.W.); zhenmengmeng@hebut.edu.cn (M.Z.); xulianfei2006@126.com (L.X.); 2019914@hebut.eud.cn (W.K.); 2School of Chemical Engineering, Hebei University of Technology, Tianjin 300401, China; shenbx@hebut.edu.cn; 3Hebei Engineering Research Center of Pollution Control in Power System, Tianjin 300401, China

**Keywords:** microalgae, pyrolysis, activated carbons, electrochemical capacitors, nitrogen self-doping

## Abstract

Pyrolysis and activation processes are important pathways to utilize residues after lipid extraction from microalgae in a high-value way. The obtained microalgae-based nitrogen-doped activated carbon has excellent electrochemical performance. It has the advantage of nitrogen self-doping using high elemental nitrogen in microalgae. In this study, two kinds of microalgae, *Nanochloropsis* and *Chlorella*, were used as feedstock for lipid extraction. The microalgae residue was firstly pyrolyzed at 500 °C to obtain biochar. Then, nitrogen-doped activated carbons were synthesized at an activation temperature of 700–900 °C with different ratios of biochar and KOH (1:1, 1:2, and 1:4). The obtained carbon materials presented rich nitrogen functional groups, including quaternary-N, pyridine-N-oxide, pyrrolic-N, and pyridinic-N. The nitrogen content of microalgae-based activated carbon material was up to 2.62%. The obtained materials had a specific surface area of up to 3186 m^2^/g and a pore volume in the range of 0.78–1.54 cm^3^/g. The microporous pore sizes of these materials were distributed at around 0.4 nm. Through electrochemical testing such as cyclic voltammetry and galvanostatic charge–discharge of materials, the materials exhibited good reversibility and high charge–discharge efficiency. The sample, sourced from microalgae *Chlorella* residue at activation conditions of 700 °C and biochar/KOH = 1:4, exhibited excellent endurance of 94.1% over 5000 cycles at 2 A/g. Its high specific capacitance was 432 F/g at 1 A/g.

## 1. Introduction

To address overconsumption of fossil resources, increasing levels of greenhouse gases in the atmosphere and the rise in the Earth’s temperature, many renewable energy technologies had been developed [1]. Microalgae are potential biomass feedstock for liquid biofuel production and industrially important by-products. Lipids in microalgae are generally used for biodiesel and liquid hydrocarbons via transesterification and hydrogenation/cracking processes. However, there are large amounts of defatted microalgae residue after extraction of lipids, which need further utilization with high added value [2]. Pyrolysis of microalgae residue can produce biochar, which can be further upgraded into activated carbon materials. Most of the pyrolysis reactions for microalgae and their defatted residue occurred at 400–550 °C with inert environment [3]. Biochar was activated by activators such as KOH and KMnO_4_ into carbon materials that can be used as supercapacitors [4,5].

Supercapacitors can be divided into three categories including electric double-layer capacitors (EDLCs), pseudocapacitors, and hybrid supercapacitors [6]. Although EDLCs have high electrical conductivity, thermal stability, and chemical stability, their specific capacitance (<400 F g^−1^) is generally lower [7]. The theoretical capacitance of pseudocapacitors (>1000 F g^−1^) is usually higher than that of EDLCs [8,9]. Pseudocapacitors generate and store electrical energy through a rapid reversible redox reaction at surface/near surface to obtain capacitance formed by metal oxides materials [10]. But attributed to agglomeration of particles, strain accumulations during charge–discharge processes, and low conductivity of metal oxides, pseudocapacitors’ practical performance was poor [11,12,13]. Therefore, these two energy storage mechanisms are combined by introducing heteroatoms into carbon materials, adjusting porosity of materials, or synthesizing a composite of carbon materials and metal oxides [14]. Heteroatom doping can significantly alter the elemental composition of carbon materials. Currently, heteroatom doping of carbon materials has been mainly studied by doping Nitrogen, Boron, Sulfur, and Platinum atoms [15,16,17,18]. Heteroatom doping, especially N doping, had shown a clear promise for high-performance supercapacitors and played an important role in carbon material doping [19]. For example, nitrogen- and oxygen-co-doped hierarchically porous carbon material from carbonization/activation of algae showed a high specific capacitance of 201 F/g at 1 A/g. In the meantime, its capacitance retention ratio was 61% at 100 A/g, and its capacitance loss was 9% after 10,000 cycles [20]. Nitrogen- and oxygen-doped hierarchically porous carbons from algae *Enteromopha prolifera* had a high capacitance of 234 F/g at 0.5 A/g and good cycling stability of 95% after 2000 cycles. Unique hierarchical nanopores in the carbon structure provided active sites for enhancing the diffusion rate of ions. These carbons had natural heteroatoms from algae-contributing pseudocapacitance [21]. One fact that should not be overlooked is that after the extraction of lipids from microalgae, the nitrogen content of the microalgal residue (5–12%) is higher than that of the raw microalgae. Therefore, it is necessary to research nitrogen distribution during pyrolysis and activation processes and its effect on pseudocapacitance activity.

Nitrogen atoms introduced into carbon materials could be divided into chemical nitrogen and structural nitrogen [22]. In general, chemical nitrogen was found as nitrogen functional groups on the surface of materials, such as amino groups and nitroso groups. These nitrogen functional groups improved the surface hydrophilic properties of porous carbon materials. It likewise increased the compatibility of electrolytes and electrodes. Structural nitrogen is doped directly into the carbon skeleton and bonded to carbon atoms. There are four main types of structural nitrogen, including graphitic-N, pyridinic-N, pyrrolic-N, and quaternary-N, and their function vary. Pyridinic-N and pyrrolic-N were effective for improving pseudocapacitance activity [23], while graphitic-N promoted the rapid transference of electrons in the carbon lattice, and quaternary-N improved the wettability between the material and the electrolyte [24]. During pyrolysis, there are different characteristics of migration and transformation for these nitrogen elements [25]. For example, high reaction pressures promoted the formation of pyrrolic-N and quaternary-N compounds in bio-oil and decreased pyridinic-N and nitrile-N compounds, while pyrrolic-N in biochars were transformed into pyridinic-N and quaternary-N at elevated pressures [26]. The step pyrolysis of N-rich industrial biowastes demonstrated that NH_3_-N was the main NO_x_ precursor in gaseous products at lower temperatures, which was due to the decomposition of labile amide-N/inorganic-N in fuels, while two-step pyrolysis with low heating rates minimized NO_x_ precursor yield by 36–43% with a greater impact on HCN-N (75–85%) than NH_3_-N (9–37%) [27]. Moreover, the distribution of nitrogen in biochars was also different in various reaction conditions [28,29]. For example, KOH dosage on the surface morphology, internal structure, and electrochemical performance of biomass-based nitrogen-doped carbon materials had significant influence. Graphitic-N (54.1–68.2 at %) accounted for a large proportion in structural nitrogen. However, with the increase in the mass ratio of KOH/porous carbon, the relative content of graphitic-N declined [30]. In the N-enriched pyrolysis of bamboo at the temperature of 400–800 °C with the reaction environment of NH_3_, pyrolysis temperature apparently decreased the relative content of pyridinic-N, while the relative content of pyrrolic-N, quaternary-N, and pyridone-N-oxide increased gradually [31]. However, there is little information about the role of elemental nitrogen in improving the electrochemical properties for microalgae-based carbon materials.

In this study, pyrolysis and activation of microalgae residue have been conducted at 500–900 °C under nitrogen flow. The characteristics of microalgae-based N-doped carbon materials have been explored to reveal the N-type distribution and its effect on pseudocapacitance activity of carbon-based materials. Finally, the electrochemical performance of microalgae-based N-doped activated carbon materials (MNAC) has been evaluated in a three-electrode system and two-electrode system with electrolyte of 6 M KOH. The items tested were cyclic voltammetry (CV), single and multiple times of galvanostatic charge–discharge cycling (GCD), and electrochemical impedance spectroscopy (EIS).

## 2. Results and Conclusions

### 2.1. Porosity of Microalgae-Based N-Doped Activated Carbon

The porosity of MNAC samples was examined through N_2_ physisorption under the condition of 77 K liquid nitrogen. In Figure 1a,b, the N_2_ adsorption isotherms of MNAC were isotherms of Type I. It indicated that MNAC were typical microporous materials [32]. The adsorption and desorption isotherms had typical hysteresis loops (Type H4). In the N_2_ isotherms, the quantity adsorbed of N_2_ rose rapidly at low relative pressures (P/P_0_ < 0.3). As the relative pressure gradually increased, the N_2_ isotherms were almost platform, indicating that the pore size distribution of MNAC samples was dominated by micropores, which was also observed in Figure 1c,d. It is worth noting that the two samples, N-800-1 and C-800-1, both with an activating agent ratio of 1:1, exhibited larger hysteresis loops, which was attributed to the smaller proportion of KOH increasing the proportion of mesopores, and a higher proportion of KOH easily led to the collapse of pores.

KOH activation could effectively enlarge the surface areas and pore volumes of carbon materials [32]. 6KOH + 2C → 2K + 3H_2_ + 2K_2_CO_3_ [33] is a conventional mechanism of carbon activation by KOH. In this reaction, potassium compounds as chemical activators etched the carbon structure by the redox reactions and then created a network of holes [34,35]. Small molecules from the decomposition of K_2_CO_3_ also created micropores. The prepared metal K was effectively intercalating into the carbon lattice of the carbon matrix during the activation process [36]. After washing off potassium ions and other potassium compounds with deionized water, the swollen carbon lattice would not be restored to its previous nonporous structure. Hence, a large surface area and pore volumes were created. In Figure 1c,d, the pore size distribution showed that the majority of pores were micropores (<2 nm), and the very few pores were mesopores (2–50 nm) and macropores (>50 nm). Meanwhile, Figure 1e,f presented that the micropore size distribution of the MNAC samples from microalgae *Nanochloropsis* and *Chlorella* all showed a peak at around 0.4 nm. All these results indicated that micropores were dominated in MNAC, which made a key contribution to the storage of charge in electrodes of EDLCs [37]. Such a pore size distribution was more conducive to the storage of charge and the transport of ions in electrochemical reactions. This is because, under low charge density, micropores were the main contributors to capacitors [13]. The micropores were mainly used for charge storage, trapping ions, and increasing capacitance, but they also impeded ion diffusion; the mesopores were responsible for ion transport and electron conduction, reducing the resistance to ion diffusion and providing a fast channel for ion diffusion [38]. However, the pore size distribution had a great impact on the specific capacitance of these carbon materials [39], which was explained in detail in Section 3.

The specific surface area (S_bet_), total pore volume (V_t_), micropore surface area (S_mic_), micropore pore volume (V_mic_), and other parameters measured from N_2_ adsorption and desorption isotherms are shown in Table 1. It was clear that the S_bet_ and V_t_ of MNAC materials are much higher than those of biochar samples N-500 and C-500 in Table 1. Among these MNAC materials, the samples of N-900-2 and C-900-2 had the highest S_bet_ (3186.74 m^2^/g and 2815.04 m^2^/g) and V_t_ (1.54 cm^3^/g and 1.29 cm^3^/g). In Appendix A, it is clear that at, the same ratio of biochar to KOH, the S_bet_ of the MNAC rose with the increase in the activation temperature. When activation temperature was the same, the specific surface areas firstly increased and then decreased with the ratio of biochar to KOH. With the increase in temperature and the ratio of biochar to KOH, the proportion of microporous areas gradually decreases. This indicated that part of the pore structure of carbon materials shifted from micropores to mesopores. When combined with the data from the subsequent electrochemical tests, there was more than a simple linear relationship between the *C*_m_ and the S_BET_ of these carbon materials. Other research had shown that the depth of the pores also determined the amount of charge accumulated on the inner wall of the pores [13], thus affecting the specific capacitance of carbon materials. When the size of the micropores matched the size of the electrolyte ions, it resulted in high transfer efficiency [38,40]. Thus, by combining with the data in Table 1 and the results of electrochemical testing in Section 3.4, it was found that the two samples with the highest specific capacitance, N-700-2 and C-800-2, had a micropore volume percentage of 78%, and their average pore diameters D_d_ were around 3.5 nm. This finding illustrated that, in order to achieve superior electrochemical properties, it was necessary to increase the specific surface area of the materials, apart from paying attention to factors such as the pore size and pore size distribution.

### 2.2. Morphology of Biochar and Microalgae-Based N-Doped Activated Carbon

Through the activation process, MNAC with high specific surface area and porous structure were obtained. To visualize biochar and MNAC, SEM images of some typical samples were obtained. The SEM of biochar (C-500 and N-500) and activated carbon (C-800-2 and N-800-2) are shown in Figure 2. In Figure 2a,b, at relatively low magnification, C-500 and N-500 had some rough formations similar to sponge structures, obvious cracks, and large pores. The pores of Figure 2b present a shape similar to the leaf veins. Compared to cylindrical pores, such shaped pores had the advantage of amplifying the specific surface area of the material and the contact area between the material and the electrolyte. Additionally, in Figure 2d,e, the samples C-800-2 and N-800-2 had a rough surface and irregular pore structures, indicating a high surface area. In the higher magnification images, as shown in Figure 2e,g, there were no visible pores on the surfaces of the biochar C-500 and N-500. However, compared to the biochar C-500 and N-500, the MNAC were etched by the alkaline activating agent KOH and presented densely packed pores with different pore sizes, especially at high magnification in Figure 2f,h.

### 2.3. Elemental Composition of Carbon Materials and Valence States of Nitrogen Atom

In Figure 3, The XPS survey spectra displayed elements O, N, and C with energy at around 533, 401, and 284.8 eV. The content of each element corresponding to each sample is listed in Table 2. Since nitrogen of microalgae was mainly concentrated in proteins, *Chlorella* with more protein content also presented more nitrogen content. In the XPS analysis, the surface nitrogen content retained by biochar (N-500 and C-500) was also high. By observing the change in nitrogen content of the materials, it could be seen that after the temperature increased, the content of nitrogen decreased significantly. This may be due to the fact that, during pyrolysis and activation processes, the higher the temperature, the more unstable nitrogen became, and detached from the carbon backbone in other forms (such as HCN and NH_3_). For environmental purposes, no additional source of nitrogen was added, but rather nitrogen in the microalgae was self-doped. The nitrogen in the MNAC could still improve the wettability of the interface between electrodes and the electrolyte, in addition to increasing the pseudocapacitance. The electrochemical properties of MNAC were presumed to be affected by a combination of surface area, pore size distribution, nitrogen content, and nitrogen doping efficiency. In Table 1, the process of activation led to a decrease in nitrogen of the materials and an increase in the specific surface area of the materials. It was therefore important to find experimental conditions that maximize the contribution of both factors at the same time.

The XPS spectrum was fitted to four or three peaks representing different valence states of elements. Figure 4 (Appendix A) shows that the N-containing constituents of activated carbons were mainly quaternary-N, pyrrolic-N, pyridinic-N, and pyridine-N-oxide. The results showed the formation of rich nitrogen-containing functional groups on the carbon framework. The nitrogenous structure, especially negatively charged pyrrole-N and pyridine-N in carbon materials, provided more active sites for faradaic pseudocapacitor reactions, while quaternary-N promoted the transport of electrons in the carbon lattice [24]. Pyrrole-N is more stable than pyridine-N, and its effect on oxygen reduction is also greater. In combination with the electrochemical performance of MNAC materials prepared at activation temperature of 800 °C, it could be concluded that the actual capacitance *C*_m_ of the MNAC varied with the total content of pyrrolic-N and pyridinic-N in the MNAC materials. The reason may be due to the larger binding energy of pyrrolic-N and pyridinic-N with ions in the electrolyte, such as the potassium ions in the KOH electrolyte in this experiment [41]. Even for a given surface area of electrodes, larger binding energy allowed a larger number of ions to be accommodated on the electrode surface, thus contributing to higher capacitance. Pyridine-N-oxide and quaternary-N are positively charged, facilitating the transfer of electrons through the carbon and improving capacitive performance under high current loads [42]. As shown in Table 3, N-Q (quaternary-N) was more stable under high-temperature pyrolysis relative to N-5 (pyrrolic-N) and N-6 (pyridinic-N). Pyridine-N-oxide and quaternary-N also provided excellent capacitance retention, while the presence of quaternary-N could enhance the electron transfer and conductivity of carbon materials, resulting in enhanced capacitance performance [10].

### 2.4. Electrochemical Performance of Microalgae-Based N-Doped Activated Carbon

#### 2.4.1. Electrochemical Performance of the Materials in the Three-Electrode System

Cyclic voltammetry curves of the samples N-900-2 and C-900-2 with various scan rates are symmetrically quasi-rectangular in Figure 5a,b. In addition, it was observed that the CV curves remained as quasi-rectangular at scan rates of 5 mV s^−1^ and 10 mV s^−1^, which suggested the typical EDLC behavior [43]. CV curves also presented a slight distortion, which reflected the combination of EDLC and pseudocapacitors related to nitrogen-containing functional groups. The introduction of nitrogen-containing functional groups on the surface of the material not only enhances the electrochemical activity of the electrode, but it also increases the hydrophilic polar sites to enhance the wettability of the electrolyte. CV curves for other samples were similar in shape to that of the samples C-900-2 and N-900-2. The shapes of the curves were closer to the rectangle after the activation temperature increased (Appendix A). As shown in Figure 5c,d, the specific capacitance (*C*_p_) measurements were calculated under the condition of sweep speed of 100 mV s^−1^, and it was found that when the activation temperature was 900 °C and the ratio of biochar to KOH was 1:2, the MNAC materials sourced from the two microalgae had the maximum value for *C*_p_, respectively.

Galvanostatic charge–discharge (GCD) cycling experiments provided the best representation of the real operation of capacitors, investigated the electrode performance, and demonstrated the characteristics of energy and power [14]. The official specific capacitance, *C*_m_ was often measured using the GCD cycling test, a test that was consistent with the actual charge–discharge of capacitors, while the specific capacitance *C*_p_ obtained from integration of the CV curve was only used as a reference. In Figure 6, the GCD curves of the MNAC materials were approximated by an isosceles triangle with excellent symmetry at various current densities, showing reversibility and high charge–discharge efficiency. Comparatively, the curves of sample N-700-2 and sample C-700-4 presented a larger area of closed curves and longer time intervals of discharging, indicating the superior electrochemical activity and higher accessible capacitance. The curves in Figure 6 exhibited the process of charging first and then discharging. For accurate calculation, the discharge processes were chosen to calculate the specific capacitance in Table 4. In Figure 6c,d, the GCD curves of the electrodes were measured at different current densities from 0.5 to 10 A g^−1^. The rate performance of samples was tested by calculation in Appendix A. The capacitance retention of samples was mostly above 80% (Appendix A) when the current density was increased to 10 A g^−1^. In combination with the results of XPS analysis, it was found that among the twenty samples tested for XPS, the two samples N-800-1 and C-700-4 with the highest total nitrogen content were also the samples with the highest specific capacitance *C*_m_ of 403.30 and 432.33 F g^−1^, respectively. This showed that there was a certain relationship between *C*_m_ and the nitrogen content of materials. It could be concluded that in MNAC capacitors, nitrogen was successfully doped into the carbon skeleton and enhanced the generation of pseudocapacitance, resulting in capacitors with higher *C*_m_.

Electrochemical impedance spectroscopy (EIS) was performed to analyze the kinetic behavior of electrochemical reaction of MNAC [44]. The Nyquist plots of all MNAC electrodes are shown in Appendix A. All Nyquist plots were similar in shape, consisting of a small diameter semicircle in the high-frequency region and an oblique line in the low-frequency region. In the Nyquist plots line, the more the line in the high-frequency region was perpendicular to the real axis (X-axis), the closer the capacitor was to an ideal electric double-layer capacitor (EDLC). As shown in Figure 7, in the equivalent circuit for the impedance spectra of the electrodes, *R*_s_ is the electrolytic solution resistance, *C*_d_ is the contact capacitance, *R*_ct_ is the charge–transfer resistance, and *W* is a Warburg diffusion element attributable to the diffusion of ions [45]. The simulation results of the equivalent circuit elements from EIS data for all of the MNAC electrodes using ZView 3.5.0.10 software are shown in Appendix A. It was well known that the semicircle in Nyquist plots corresponded to charge–transfer resistance (*R*_ct_). In Appendix A, it shows that N-700-4, N-800-2, N-900-2, C-800-2, and C-900-2 had lower *R*_ct_ of 0.094, 0.083, 0.090, 0.142, and 0.179 Ω, respectively, facilitating faster charge transfer and easier access of electrolyte to MNAC electrodes. Combined with the data in Table 1, the lower *R*_ct_ values of MNAC electrodes were mainly attributed to their higher S_BET_, abundant mesopores and micropores. In the middle frequency range, the lines represented the ions diffusion resistance (*W*) and approached 45° (Warburg line), reflecting the process of ion transport/diffusion between the electrode/electrolyte [46]. In Appendix A, the higher values of *W* for MNAC electrodes (such as the sample C-900-4) indicated that the diffusion of ions in the electrode was significantly hindered [47]. Smaller values of *R*_sum_ facilitated faster charging and discharging of the electrodes at high current densities and increased the capacitance of MNAC [32,47].

#### 2.4.2. Electrochemical Performance of the Materials in the Two-Electrode System

In the previous electrochemical evaluation, sample C-700-4 presented an outstanding electrochemical performance with a specific capacitance *C*_m_ of 432.33 F g^−1^ and was therefore selected for testing the practical application of MNAC electrode materials. A symmetric supercapacitor based on C-700-4 was tested in a two-electrode system in 6 M KOH electrolyte. In Figure 8b, the GCD curves of the C-700-4 supercapacitor at current densities from 0.25 to 20 A g^−1^ are approximately isosceles triangular in shape, indicating ideal supercapacitor behavior and excellent electrochemical reversibility. Figure 8a presents the relationship between the current density and specific capacitance (*C*_s_) of C-700-4, from which it can be seen that the specific capacitance decreases with the increase in the current density, and the specific capacitance of C-700-4 reaches as high as 286 F g^−1^ at a current density of 0.5 A g^−1^, and its specific capacitance is 132 F g ^−1^ when its current density is increased from 0.5 to 20 A g^−1^, and the final capacitance retention is 46.2%.

To test the cycling stability of C-700-4 electrode, a galvanostatic charge–discharge test was conducted at a current density of 2 A g^−1^, between 0 and 1 V for 5000 cycles. In Figure 8c, the capacitance retention for the sample C-700-4 was highly up to 94.1% over 5000 cycles. It confirmed that the sample C-700-4 had an excellent and long-term stability as an electrode for supercapacitor. In Figure 8d, the energy density (*E*) of C-700-4 demonstrates a high energy density of 42.6 Wh kg^−1^ at the power density (*P*) of 250 W kg^−1^, while it retained 18.3 Wh kg^−1^ at elevated power density of 2000 W kg^−1^. In order to prove the practical outstanding working performance of this work, these supercapacitor materials prepared by other algae-based carbon materials were compared in the Ragone plot [13,48,49,50]. It was shown that the supercapacitor C-700-4 had the highest energy density and power density among the other algae-based carbon materials in recent reports.

## 3. Materials and Methods

### 3.1. Microalgae and Lipid Extraction

The powder microalgae samples (*Nanochloropsis* and *Chlorella*) used in this study were collected from Qingdao, China, with the particle size of 0.25 μm. Before each experiment, both microalgae samples were dried at 105 °C for 24 h. The extraction of lipids was conducted via organic solvent extraction method. The used organic solvents were CH_2_Cl_2_ (analytical reagent, >99%) and CH_3_OH (analytical reagent, >99%), which were mixed into 300 mL of organic solution according to the volume ratio of 1:2. About 15 g of microalgae powder was mixed with the organic solvent and extracted in an ultrasonic machine for 120 min. The temperature of the extraction system was maintained at 20 °C through a water bath. After extraction, the resulting mixture was centrifuged for 12 min at 6500 rpm to separate solid product from organic solvent, and the solid product was washed with deionized water and centrifuged several times to remove any residual organic solvent. Finally, the solid product was dried at 80 °C in an electrothermal blowing dry box to obtain dried microalgae residue. About 10.5 g of microalgae residue was obtained after lipid extraction of 15 g of microalgae powder. The yield was approximately 70%. The composition of microalgae *Nanochloropsis* and *Chlorella* and their profiling of amino acids are shown in Appendix A. The lipid content of microalgae was determined by the method of organic solvent extraction. The organic solvents were the previously mentioned methanol and dichloromethane. The mass balance method was used to calculate carbohydrate content. The premise is that the sum content of protein, lipid, water, and ash in microalgae was 100%. Microalgae were completely burned in a muffle furnace to determine their ash content. The elementary analysis (C, H, S, and N) of microalgae *Nanochloropsis* and *Chlorella* was performed by an elemental analyzer (Thermo Scientific FlashSmart, Stoney Creek, NC, USA). The amino acid composition of microalgae was determined by an amino acid analyzer (MembraPure A300, Berlin, Germany).

### 3.2. Pyrolysis of Microalgae and Activation of Biochar

A scheme of the pyrolysis and activation setup is shown in Figure 9. The microalgae residue after extraction of lipids was put into a cylindrical graphite crucible (50 × 40 × 50 mm) with a lid. In order to obtain biochar, the microalgae residue was firstly pyrolyzed for 1 h in a furnace at 500 °C under N_2_ flow. The flow rate of N_2_ flow was 400 mL/min. The heating rate of the furnace was 5 °C/min. After pyrolysis with 10.5 g of microalgae residue, about 3.2 g of biochar could be obtained, indicating that the yield of biochar was approximately 30%. Then, the obtained biochar was mixed with the activating agent potassium hydroxide (KOH) in ratios of 1:1, 1:2, and 1:4 in the form of powder. The mixture was put into a crucible of the same size. Finally, the crucible was put back into the furnace and activated at 700 °C, 800 °C, or 900 °C for 1 h. Both the nitrogen flow rate (400 mL/min) and the heating rate (5 °C/min) were the same as the pyrolysis process. The obtained activated carbons were mixed with excess of 1 mol/L HCl, stirred at 25 °C for 12 h to make the acid and base react completely, and rinsed with distilled deionized water until the pH reached neutrality. Afterwards, it was dried at 80 °C for 12 h in an electrothermal blowing dry box. Finally, the resulting solid was microalgae-based N-doped activated carbon material (MNAC). After activation with 3.2 g of biochar, about 1.5 g of MNAC could be obtained, indicating that the yield of MNAC was approximately 47%.

The MNAC samples at different experimental conditions were denoted as N-T-p or C-T-p. The first letter N represented the microalgae *Nanochloropsis*, and the letter C represented microalgae *Chlorella*. The second number T represented the activated temperature, and the third number p represented the different ratio of KOH. For example, the named sample of N-700-2 means the activated carbon sourced from pyrolysis and activation of microalgae *Nanochloropsis* with activation temperature of 700 °C and the ratio of biochar/KOH of 1:2, while the named sample of C-900-1 means the activated carbon sourced from pyrolysis and activation of microalgae *Chlorella* with activation temperature of 900 °C and the ratio of biochar/KOH of 1:1.

### 3.3. Characterization of Biochar and Microalgae-Based N-Doped Activated Carbon

Under the condition of 77 K liquid nitrogen, the samples were tested for nitrogen adsorption and desorption with an N_2_ adsorption–desorption analyzer (Micromeritics ASAP 2460, Norcross, GA, USA). The micropore volume and micropore surface areas were determined by the *t*-plot method. The total specific surface areas of the materials were obtained by the BET method. At a relative pressure (P/P_0_) of 0.95, the adsorption amount of nitrogen is used to determine the total pore volume. In order to study the elemental composition and relative content of the surface of the materials and the valence state of nitrogen atoms, X-ray photoelectron spectroscopy (XPS) (Thermo Scientific K-Alpha, Stoney Creek, NC, USA) analysis was carried out. An electron microscope scanning electron microscopy (SEM) (TESCAN MIRA LMS, Brno, Czech Republic) was used to observe the morphology of the materials with an acceleration voltage of 3 kV.

### 3.4. Electrochemical Measurements of Microalgae-Based N-Doped Activated Carbon

The electrochemical performance of MNAC prepared by the pyrolysis and activation process was evaluated. The composition of the electrode material was 80 wt % of MNAC, 10 wt % of PTEP, and 10 wt % of acetylene black. After being added with 5 mL of anhydrous ethanol, the electrode material was stirred for 3 h at a water bath temperature of 45 °C. The evaporated solid powder was rolled out into flakes and dried for more than a day. The electrode flakes were cut into 1 cm^2^ and then pressed against the strips of nickel foam at a pressure of 10 MPa.

The electrochemical measurements were performed in the beaker-type three-electrode cell using 6 M KOH electrolyte solution at an ambient temperature. The beaker-type three-electrode cell was equipped with electrode materials prepared from the MNAC samples on nickel foam as the working electrode, an HgO electrode as the reference electrode, and a 1 cm^2^ sheet of platinum as the counter electrode.

Cyclic voltammetry (CV) measurements were performed using a Model 700E Series Electrochemical Analyzer (CH Instruments, Shanghai, China) at different scan rates (5 mV/s, 10 mV/s, 20 mV/s, 50 mV/s, and 100 mV/s). The potential window was from 0 V to −1 V. The specific gravimetric capacitance of the three-electrode cell, *C*_p_ (F g^−1^), was calculated from the area of the voltammograms using Equation (1):(1)$$C_{p}=\frac{A}{m\cdot k\cdot \Delta V}$$
where *A* (C) is the integral area of the curves, *m* (g) is the mass of carbon materials in each electrode, *k* (V s^−1^) is the scan rate, and Δ*V* (V) is the potential window.

Galvanostatic charge–discharge (GCD) measurements were performed using a Model 700E Series Electrochemical Analyzer (CH Instruments, China, Shanghai) in a range of constant current densities (0.5 A g^−1^, 1 A g^−1^, 2 A g^−1^, 5 A g^−1^, and 10 A g^−1^). The potential window was from −1 V to 0 V. *C*_m_ (F g^−1^), and the specific gravimetric capacitance was calculated by the discharge curves using Equation (2):(2)$$C_{m}=\frac{I\Delta t}{m\Delta U}$$
where *I* (A) is the current densities, Δ*t* (s) is the discharge time, *m* (g) is the mass of the active substance, and Δ*U* (V) is the potential window.

Electrochemical impedance spectroscopy (EIS) was also performed using a Model 700E Series Electrochemical Analyzer (CH Instruments, China, Shanghai). This was to study the internal resistance of the electrode materials and the impedance behavior between the electrode materials and the electrolyte at the open-circuit voltage of the samples. The test condition was a frequency range of 10 mHz–100 kHz and AC current amplitude of 5 mV.

To investigate practical application performance of the material, electrode material prepared from the MNAC samples was tested in the beaker-type symmetrical two-electrode cell using 6 M KOH electrolyte solution. The specific capacitance *C*_s_ (F g^−1^) of the two-electrode system was calculated using Equation (3):(3)$$C_{s}=\frac{2I\Delta t}{m\Delta U}$$
where *I* (A) is the current density, Δ*t* (s) is the discharge time, *m* (g) is the mass of the active substance, and Δ*U* (V) is the voltage change of the system.

After calculating the value of *C*_s_, the energy density *E* (Wh kg^−1^) was calculated using Equation (4):(4)$$E=\frac{1}{2}C_{s}\Delta U^{2}\times \left(1000/3600\right)$$
where Δ*U* is the operating voltage of the system (V).

The power density *P* (W kg^−1^) was calculated using Equation (5):(5)P=3600E/Δt
where Δ*t* is the discharge time (s).

## 4. Conclusions

In this study, porous N-doped activated carbon materials were successfully obtained using microalgae residue after extraction of lipids as a precursor with the coupled processes of pyrolysis (500 °C) and activation (700–900 °C) with KOH. The choice of nitrogen-rich microalgae precursors allowed the final carbon materials to be successfully self-doped with nitrogen (0.27–2.62%). The microalgae-based N-doped activated carbons combined a high specific surface area up to 3184 m^2^/g and the high proportion of microporous, providing space for ion storage and fast lanes for transportation, enhancing the outstanding electrochemical performance of MNAC. The N-doped activated carbon material with microalgae *Chlorella* as precursor, an activation temperature of 700 °C, and a biochar-to-KOH ratio of 1:4 had the highest specific capacitance *C*_m_ of 432.33 F g^−1^. Its cycling stability was also excellent with a capacitance retention of 94.1% at over 5000 cycles.

## Figures and Tables

**Figure 1 molecules-28-07280-f001:**
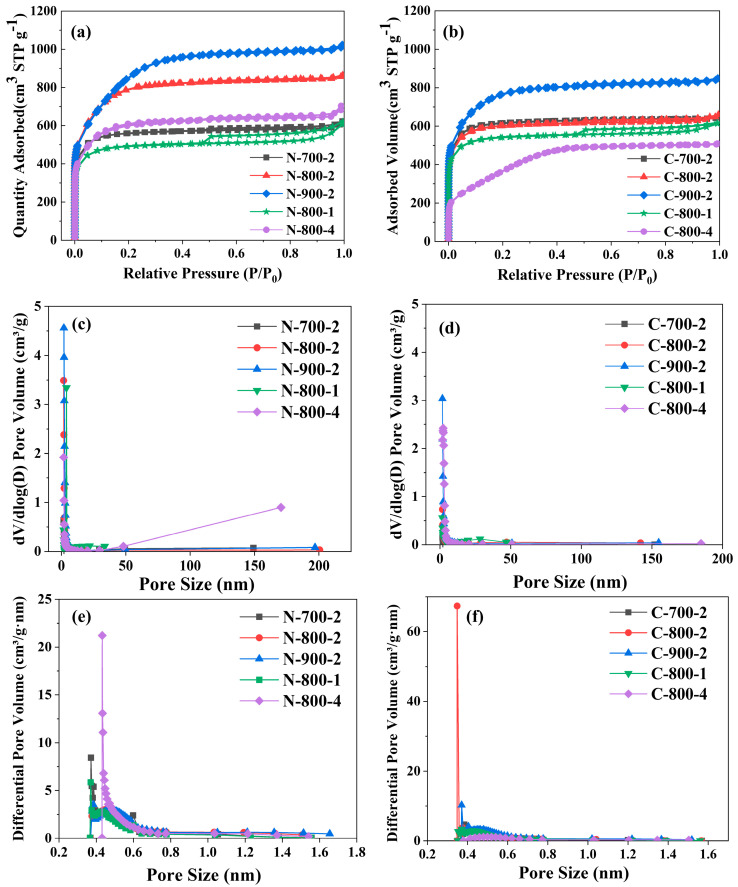
N_2_ adsorption and desorption isotherms of (**a**) *Nanochloropsis*-based MNAC; (**b**) *Chlorella*-based MNAC; (**c**) pore size distribution of *Nanochloropsis*-based MNAC; (**d**) pore size distribution of *Chlorella*-based MNAC, (**e**) micropore size distribution of *Nanochloropsis*-based MNAC; (**f**) micropore size distribution of of *Chlorella*-based MNAC.

**Figure 2 molecules-28-07280-f002:**
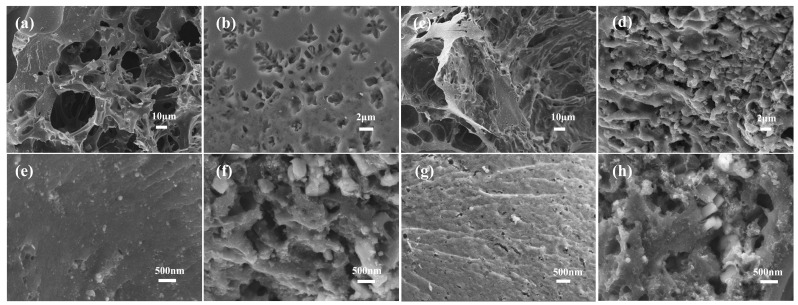
SEM image of the samples of (**a**,**e**) biochar C-500, (**b**,**f**) C-800-2, (**c**,**g**) biochar N-500, and (**d**,**h**) N-800-2.

**Figure 3 molecules-28-07280-f003:**
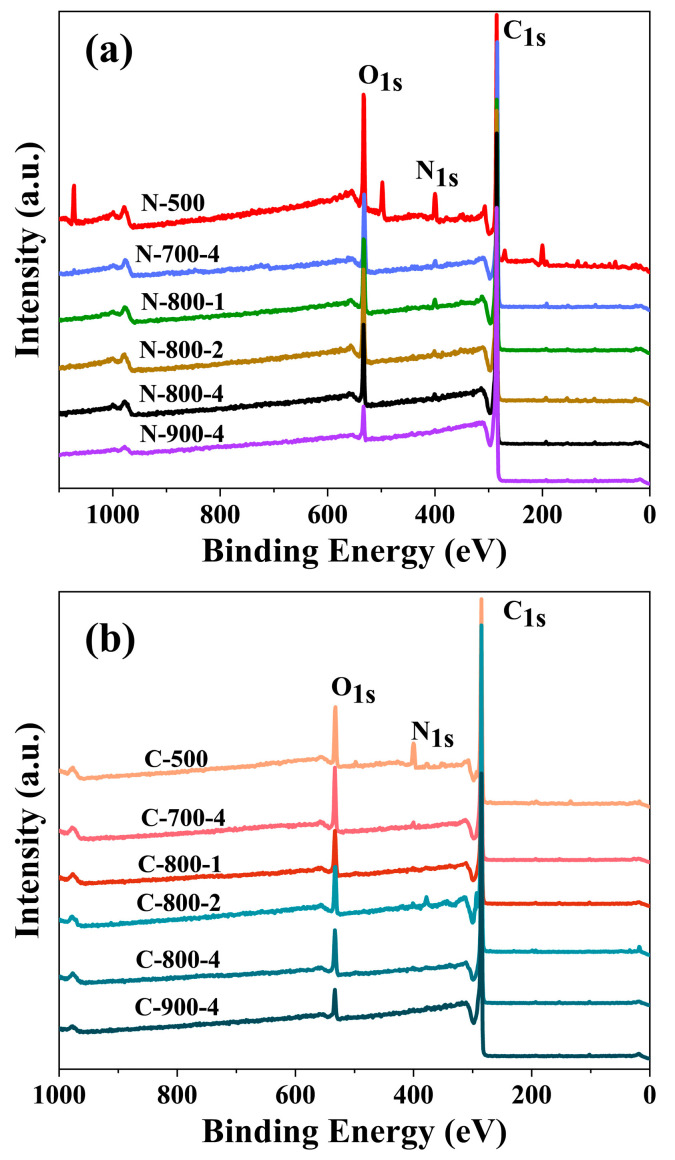
The XPS survey spectra of carbon materials (**a**) N-500, N-700-4, N-800-1, N-800-4, N-800-2, and N-900-4, (**b**) C-500 and C-700-4, C-800-1, C-800-4, C-800-2, and C-900-4.

**Figure 4 molecules-28-07280-f004:**
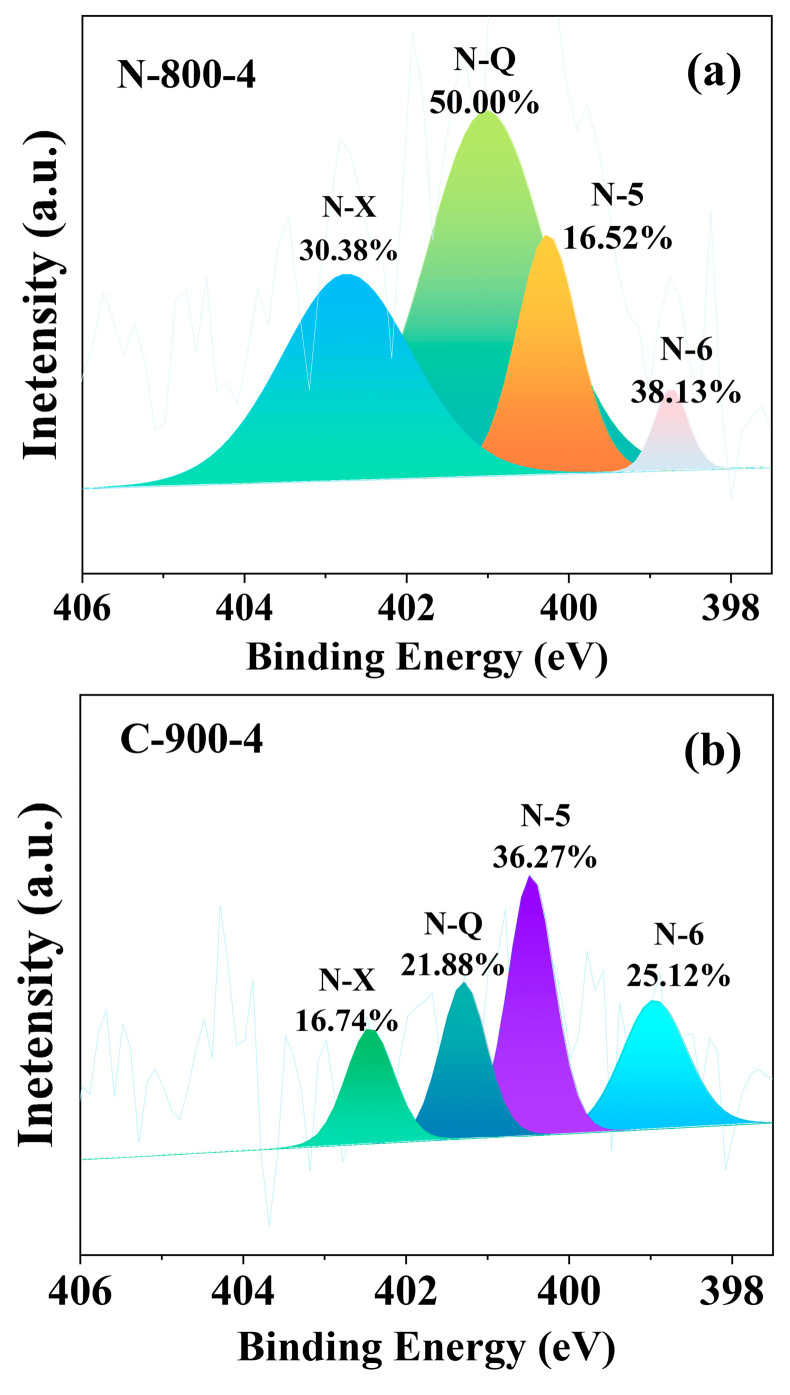
N 1 s spectra of (**a**) sample N-800-4 and (**b**) sample C-900-4.

**Figure 5 molecules-28-07280-f005:**
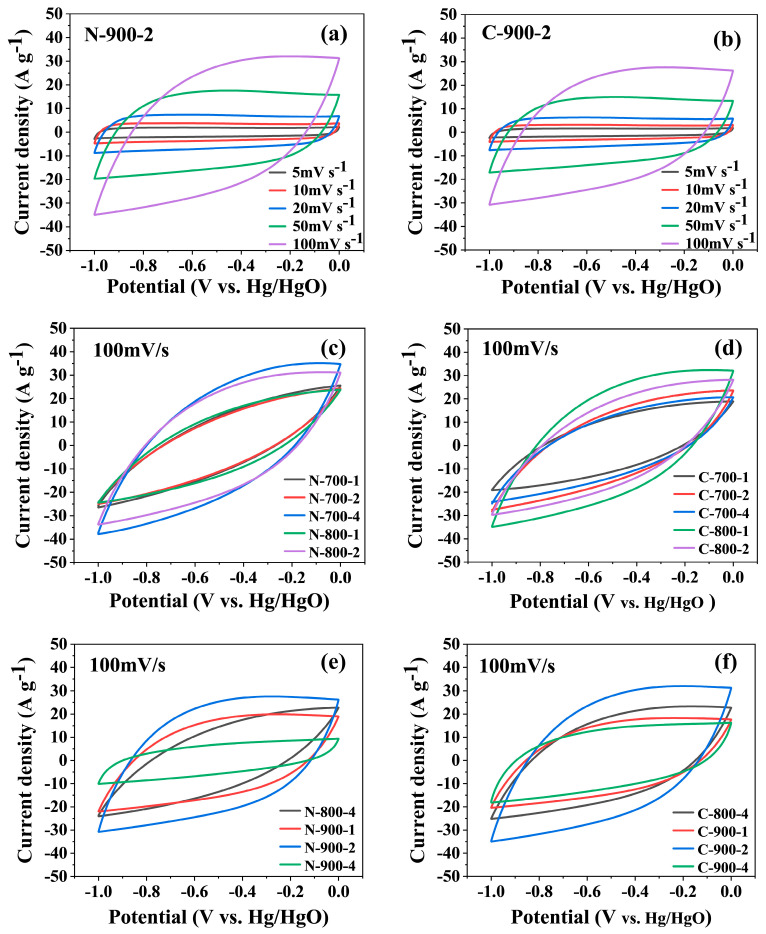
CV curves: (**a**) sample N-900-2; (**b**) sample C-900-2; (**c**,**e**) *Nanochloropsis*-based MNAC materials; (**d**,**f**) *Chlorella*-based MNAC materials.

**Figure 6 molecules-28-07280-f006:**
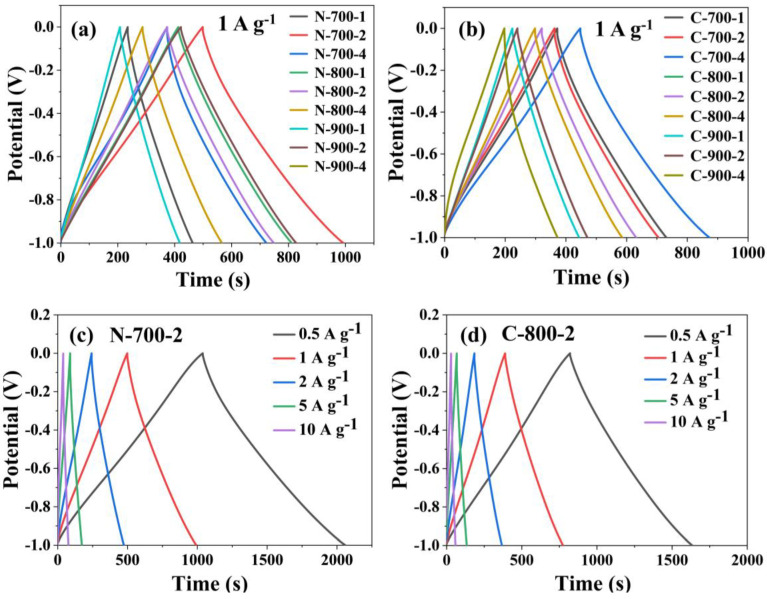
GCD curves (**a**) *Nanochloropsis*-based MNAC materials at 1 A g^−1^; (**b**) *Chlorella*-based MNAC materials at 1 A g^−1^; (**c**) N-700-2 at all current density; (**d**) N-700-2 at all current density.

**Figure 7 molecules-28-07280-f007:**
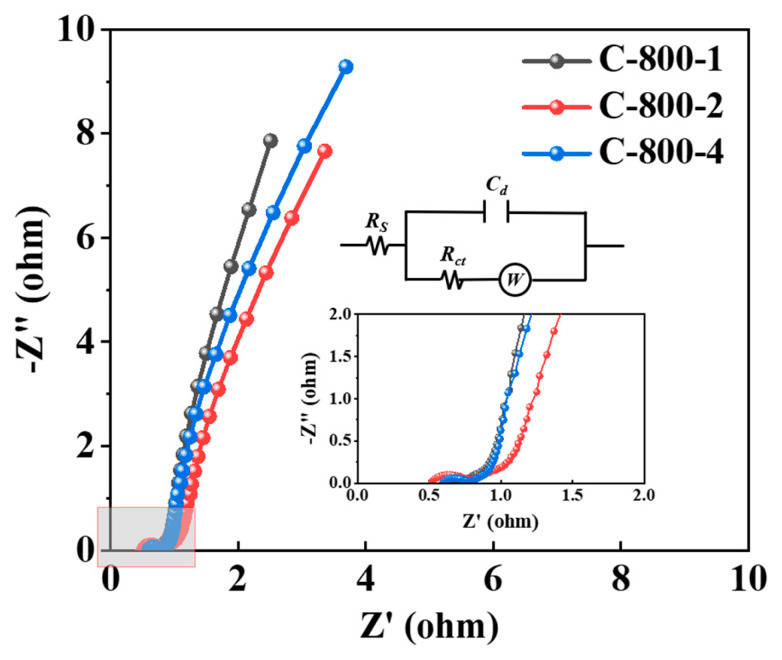
Nyquist plots of the capacitors of C-800-1, C-800-2, C-800-4 and their equivalent circuit.

**Figure 8 molecules-28-07280-f008:**
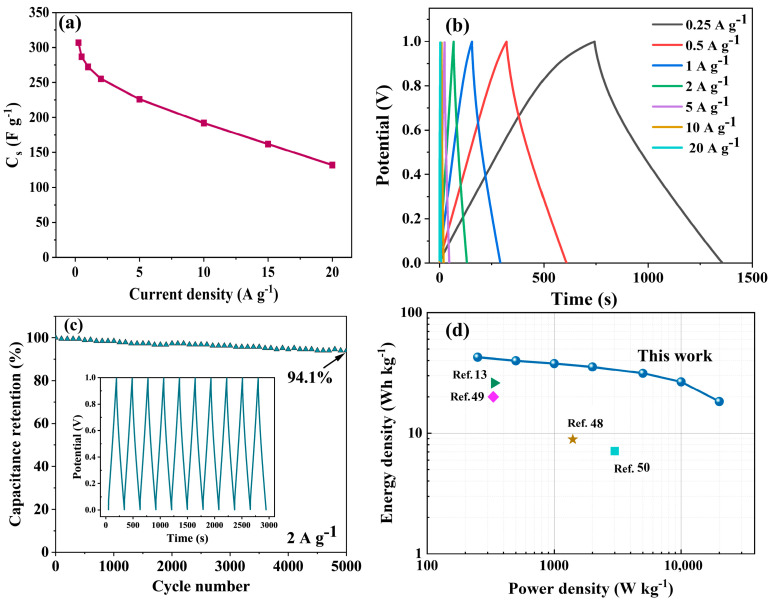
Electrochemical performance of supercapacitor based on C-700-4 tested in two-electrode system: (**a**) Specific capacitance from 0.25 to 20 A g^−1^; (**b**) GCD curves at various current densities; (**c**) The cycling performance at 2 A g^−1^; (**d**) Ragone plot [13,48,49,50].

**Figure 9 molecules-28-07280-f009:**
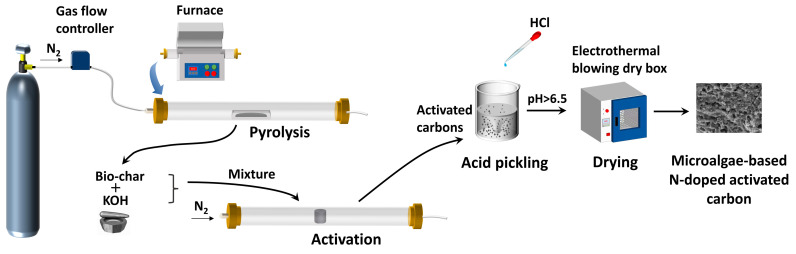
Scheme of pyrolysis and activation setup.

**Table 1 molecules-28-07280-t001:** Specific surface area and pore structure of biochar and microalgae-based N-doped activated carbon.

Microalgae Species	Sample	S_BET_ (m^2^/g)	V_t_ ^a^ (cm^3^/g)	D_d_ ^b^ (nm)	S_mic_ ^c^ (m^2^/g)	V_mic_ ^d^ (cm^3^/g)	Micropore Surface Area (%)	Micropore Volume (%)
*Nanochloropsis*	N-700-2	2134.73	0.91	3.66	1803.75	0.72	84.50	78.62
	N-900-2	3186.74	1.54	2.32	2055.13	0.76	64.49	49.47
	N-800-1	1898.70	0.86	4.42	1593.00	0.62	83.90	72.15
	N-800-2	2946.12	1.31	2.30	1745.69	0.68	59.25	52.12
	N-800-4	2223.03	1.00	3.00	1366.59	0.56	61.47	55.66
*Chlorella*	C-700-2	2397.06	0.99	2.80	2149.50	0.84	89.67	85.30
	C-900-2	2815.04	1.29	2.34	1365.55	0.54	48.51	41.79
	C-800-1	2099.36	0.92	3.90	1790.52	0.70	85.29	75.91
	C-800-2	2495.37	0.98	3.41	2028.98	0.77	81.03	78.14
	C-800-4	1317.35	0.78	2.39	624.63	0.22	47.42	28.48

^a^ Total pore volume was determined by less than 40.3 nm diameter. ^b^ Average pore diameter was calculated by BJH desorption. ^c^ Micropore area was calculated by *t*-Plot method. ^d^ Micropore volume was calculated by *t*-Plot method.

**Table 2 molecules-28-07280-t002:** Elemental content of carbon materials measured by XPS.

Sample	Element Content (at %)		
	C	O	N
N-500	56.36	14.56	6.23
N-700-4	84.89	13.05	2.05
N-800-1	85.83	11.55	2.62
N-800-4	87.14	10.77	2.09
N-800-2	85.04	13.07	1.90
N-900-4	93.66	5.11	1.23
C-500	77.69	12.65	9.38
C-700-4	85.99	12.08	1.93
C-800-1	92.11	7.64	0.25
C-800-4	86.05	12.47	1.53
C-800-2	89.90	9.83	0.27
C-900-4	93.32	5.44	1.24

**Table 3 molecules-28-07280-t003:** The content of different forms of nitrogen in carbon materials (at %).

Sample	Quaternary-N	Pyrrolic-N	Pyridinic-N	Pyridine-N-Oxide
N-500	57.28	26.21	14.56	-
N-700-4	45.80	35.56	18.46	-
N-800-1	40.00	10.48	29.85	19.66
N-800-4	50.00	16.52	3.10	30.38
N-800-2	82.84	14.88	2.92	-
N-900-4	87.57	4.68	7.75	-
C-500	28.81	34.91	36.28	-
C-700-4	35.52	16.17	36.81	11.51
C-800-1	84.23	4.48	11.29	-
C-800-4	8.22	57.38	26.88	7.52
C-800-2	43.03	22.83	14.27	19.43
C-900-4	21.88	36.27	25.12	16.74

**Table 4 molecules-28-07280-t004:** Specific capacitance of microalgae-based N-doped activated carbon at 1 A g^−1^.

Sample	*C*_m_ (F g^−1^)	Sample	*C*_m_ (F g^−1^)
N-700-1	234.94	C-700-1	368.46
N-700-2	501.87	C-700-2	351.87
N-700-4	358.01	C-700-4	432.33
N-800-1	403.30	C-800-1	318.95
N-800-2	380.74	C-800-2	391.20
N-800-4	283.45	C-800-4	293.78
N-900-1	213.69	C-900-1	224.04
N-900-2	408.85	C-900-2	233.84
N-900-4	243.43	C-900-4	178.52

## Data Availability

The data that support the findings of this study are available upon reasonable request.

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
