# Peer review of "Electrochemical Performance of Nitrogen Self-Doping Carbon Materials Prepared by Pyrolysis and Activation of Defatted Microalgae"

_molecules, 2023, doi:10.3390/molecules28217280_

Round 1
Reviewer 1 Report
It is an interesting study, and the conclusions are mostly supported by the data, which may merit publication after fully addressing the following issues.
1) Title: It would be better to replace the title with “Electrochemical capacitors based on nitrogen self-doped carbon materials derived from defatted croalgae”
2) Abstract: sentences “Microalgae-based activated carbons prepared by pyrolysis and activation process is an important pathway for high-value utilization of microalgae residue after lipid extraction” and “The surface areas of obtain materials were 1317–3186 m2 /g” need to be rewritten.
3) Abstract: A full name should be provided to replace CV and GCD.
4) Keywords: "electrochemical material" should be replaced by "electrochemical capacitors"; "activation" should be replaced by "activated carbons"
5) Introduction: some closely-related publications need to be cited to enhance the research background, e.g., A) B. Fang, F. Heuveln, F. Dias, and L. Plomp, Electric double-layer capacitor based on activated carbon material, Rare Metals 2000, 19(1), 1-10; B) B. Fang, Y. Wei, K. Suzuki and M. Kumagai, Surface modification of carbonaceous materials for EDLCs application, Electrochim. Acta 2005, 50(18), 3616-3621; C) Z. Pan, et al., Recent advances in porous carbon materials as a supercapacitor electrode, Nanomaterials 2023, 13, 1744
6) Introduction: please replace “Pyrolysis of microalgae residue can produce bio-char via pyrolysis” with “Pyrolysis of microalgae residue can produce bio-char”
7) Introduction: The sentence “Bio-char from pyrolysis is also activated into high-performance carbon materials as supercapacitors by activating agent such as KOH and KMnO4” need to be rewritten
8) Please move Figure 1, Table 1 and Table 2 from the main text to Supplementary Information.
9) Figure 2: please use the same symbols and line colors for all N-* samples, and use the same symbols and line colors for all C-* samples as well.
10) Line 177: please replace “Average pore diameterIt was calculated by BJH desorption” with “Average pore diameter was calculated by BJH desorption”
11) Line 184: Replace “were shown in Figure 3” with “are shown in Figure 3”. In addition, please change Figure 3(a)(b) to Figures 3a and 3b. Please correct similar errors throughout the manuscript.
12) Figure 3, caption: please change it to “SEM images of the samples of (a, e) bio-char C-500, (b, f) C-800-2, (c, g) bio-char N-500, and (d, h) N-800.”
13) Figure 4, Y axis unit: it should be “a. u.” rather than “a. u”
14) Figure 6, caption: please replace “Cyclic voltammetry (CV) curves” with “CV curves” because you have defined “Cyclic voltammetry” as CV previously, and you don’t need to define it again here. Please correct similar error in Figure 7, caption.
15) Author Contributions: Initials of the authors should be provided here.
16) English needs to be polished.
Please polish English significantly
Author Response
Dear reviewer, I have adopted the vast majority of your comments. I have revised it in the original and marked the modified position in red.

Reviewer 2 Report
- Text in lines 83-86 is not clear.
- Lines 360-361 it is not clear how washing with water would help removing the organic CH2Cl2,
- There is no Table 1 and Table 2 before Table 3.
- Figure 3.b shows clear patterns similar to flakes/branches/leaves; this should be properly discussed.
Author Response

(The authors gave the same response as above.)
